# Low Skeletal Muscle Mass Is a Risk Factor for Subclinical Atherosclerosis in Patients with Nonalcoholic Fatty Liver Disease

**DOI:** 10.3390/diagnostics11050854

**Published:** 2021-05-10

**Authors:** Min-Kyu Kang, Jung-Gil Park

**Affiliations:** Department of Internal Medicine, College of Medicine, Yeungnam University, Daegu 42415, Korea; kmggood111@naver.com

**Keywords:** atherosclerosis, nonalcoholic fatty liver disease, sarcopenia, cardiovascular diseases

## Abstract

Sarcopenia and nonalcoholic fatty liver disease (NAFLD) are associated with subclinical atherosclerosis. We aimed to investigate the association between low skeletal muscle mass (LSMM) and subclinical atherosclerosis in patients with NAFLD. A total of 683 patients with ultrasound-confirmed NAFLD who underwent carotid ultrasonography were enrolled retrospectively. The appendicular skeletal muscle mass divided by the body mass index was used to define LSMM. Using carotid ultrasound, increased carotid intima–media thickness (cIMT, >1 mm) and the presence of carotid plaques were measured. Of the 683 patients, 75 (11.0%) had LSMM. In multivariate analyses, LSMM was associated with increased cIMT (odds ratios (ORs) = 2.26 to 2.95, all *p* < 0.05) and carotid plaques (ORs = 2.05 to 2.90, all *p* < 0.05). The proportion of increased cIMT and carotid plaques was significantly higher in obese NAFLD patients with LSMM than in those without LSMM (33.3% vs. 17.6% for cIMT and 12.7% vs. 5.7% for carotid plaques, respectively; *p* < 0.001). Furthermore, LSMM was independently associated with increased cIMT (ORs = 2.44 to 3.30, all *p* < 0.05) and carotid plaques (ORs = 2.56 to 3.54, all *p* < 0.05) in obese NAFLD patients. LSMM is associated with subclinical atherosclerosis in patients with NAFLD.

## 1. Introduction

Nonalcoholic fatty liver disease (NAFLD) is characterized by fat accumulation in hepatic tissues and is one of the most common metabolic liver diseases. This disease presents with a diverse spectrum of progressive liver pathologies ranging from simple steatosis, steatohepatitis, and fibrosis to severe conditions such as cirrhosis and hepatocellular carcinoma [1]. In accordance with previous studies, NAFLD is considered both a multisystemic disease and a disease confined to the liver due to complex mechanisms. Furthermore, cardiovascular disease (CVD) is frequently associated with NAFLD and is a major cause of death in affected patients [2,3].

The progression of NAFLD is linked to altered systemic metabolic conditions due to chronic inflammation, which may partly contribute to the increased risk of CVD events including myocardial infarction, arrhythmia such as atrial fibrillation, and valvular heart disease [4,5,6,7]. Atherosclerosis is a chronic inflammatory process that is pivotal in causing CVD events [8]. Subclinical atherosclerosis can be diagnosed by documenting the presence of increased carotid intima–media thickness (cIMT) and carotid plaques (CPs) using carotid ultrasound; this pathology is an early predictive marker for CVD events [2,9,10].

Sarcopenia is an age-related progressive disease characterized by low muscle strength as a prerequisite, low muscle quantity or quality, and low physical performance [11]. This disease is associated with an increased risk of physical disability, mortality in the aging population, and CVD events in patients with diabetes mellitus (DM) and chronic kidney disease [11,12,13]. Sarcopenic obesity, defined as the concurrence of sarcopenia and obesity, is regarded as a more serious condition than sarcopenia or obesity alone. Sarcopenic obesity is also associated with metabolic disorders, CVD events, and NAFLD [14,15,16]. Previous studies have revealed an association between sarcopenia/sarcopenic obesity and NAFLD [17,18,19]. Although the association between sarcopenia and subclinical atherosclerosis has been investigated in patients with type 2 DM and in older populations, only one study has investigated the association between sarcopenia and atherosclerotic cardiovascular diseases in patients with NAFLD using serum tests [20]. Although there are advantages of the large national population-based cohort in this study, liver or carotid imaging widely used clinically was not available due to high cost of ultrasonographic examination. Therefore, we investigated the association between low skeletal muscle mass (LSMM) and subclinical atherosclerosis in patients with NAFLD using ultrasound.

## 2. Materials and Methods 

### 2.1. Patients

This was a retrospective, cross-sectional, single-institution study that assessed the association of LSMM with carotid atherosclerosis in patients with NAFLD. From January 2013 to November 2018, 2552 subjects who underwent carotid ultrasound at a health promotion center were enrolled.

A total of 1869 subjects were excluded based on the following criteria: (i) no evidence of fatty liver on abdominal ultrasound (*n* = 1670); (ii) positive serologic findings for hepatitis B (*n* = 137) or C (*n* = 21); and (iii) history of excessive alcohol consumption (men > 140 g/week; women > 70 g/week) (*n* = 41) (Figure 1) [21].

A total of 683 patients with NAFLD who underwent carotid ultrasonography were included in the study. Informed consent from participants was waived because of the retrospective nature of this study. The study protocol was approved by the Yeungnam University Hospital Institutional Review Board (2020-03-008).

### 2.2. Measurements of Clinical and Laboratory Data 

Anthropometric assessments, including weight, height, seated blood pressure, and waist circumference (WC), were performed and recorded by trained nurses. Data on any history of alcohol consumption and comorbidities were collected using a self-administered questionnaire. Blood sample collection and abdominal ultrasound were done after each participant completed an overnight fast of 12 h. The patients’ liver profiles, including serum aspartate aminotransferase (AST), alanine aminotransferase (ALT), γ-glutamyltransferase (GGT), albumin, and platelet count; lipid profiles, including total cholesterol (TC), high-density lipoprotein cholesterol (HDL-C), and triglyceride (TG); and glucose profiles, including fasting plasma glucose (FPG), insulin level, and homeostasis model of insulin resistance (HOMA-IR), were measured.

Based on the Asia-Pacific region criteria, obesity was defined as a body mass index (BMI) ≥ 25 kg/m^2^ [22]. Using the American Diabetes Association criteria, DM was defined as an FPG level ≥ 126 mg/dL or the use of antidiabetic medications. Hypertension was defined as (i) a seated systolic BP of ≥140 mmHg, (ii) a diastolic BP of ≥90 mmHg, or (iii) intake of any antihypertensive medications. Based on the International Diabetes Federation criteria, metabolic syndrome in Asian adults was defined as the presence of visceral obesity (WC ≥ 90 cm in men and ≥85 cm in women) plus two of the following factors: elevated TG (≥150 mg/dL), reduced HDL-C (≤40 mg/dL in men and ≤50 mg/dL in women), elevated BP (systolic BP ≥ 130 mmHg or diastolic BP ≥ 85 mmHg), or elevated FPG (≤100 mg/dL) [23]. Hyperuricemia was defined as a serum urate level greater than 6.8 mg/dL [24]. Insulin resistance (IR) was defined using the homeostasis model assessment (HOMA) method as follows: HOMA-IR = fasting insulin (IU/mL) × fasting glucose (mg/dL)/405 [25].

### 2.3. Assessment of Low Skeletal Muscle Mass 

On the same day that carotid and abdominal ultrasound were performed, the appendicular skeletal muscle mass (ASM) of the participants was measured through bioelectrical impedance analysis (BIA) using an InBody 720 body composition analyzer (Biospace, Seoul, Korea). The ASM was calculated as the sum of the lean muscle mass of the four limbs. Based on our study published previously, we presented an LSMM model defined as the ASM divided by BMI (LSMM_BMI) [26]. Based on the definition of LSMM by the Foundation for the National Institutes of Health, LSMM_BMI was defined as <0.789 in males or <0.512 in females [27].

### 2.4. Assessment of NAFLD 

Using algorithms from previous publications, the presence of fatty liver was determined by two experienced radiologists using EPIQ 5 and EPIQ 7 (Philips, Amsterdam, The Netherlands) based on the following criteria: (1) increased echogenicity of the liver parenchyma relative to that of the cortex of the kidneys, (2) deep beam attenuation, and (3) blurring of the intrahepatic vessels [4,26]. Based on the guidelines by the Asia-Pacific Working Party on Non-alcoholic Fatty Liver Disease, NAFLD was defined as the presence of a fatty liver in the absence of any of the proposed exclusion criteria [21].

### 2.5. Assessment of Carotid Atherosclerosis

The bilateral common carotid arteries (CCAs) were examined by two radiologists using EPIQ 7 (Philips, Amsterdam, The Netherlands), which is a high-resolution ultrasound with a linear transducer. The mid- and distal CCAs were scanned at the lateral longitudinal projection. The cIMT was measured at three points: (i) the distal wall of the mid-CCA, (ii) the distal CCA, and (iii) approximately 1 cm proximal to the carotid bulb. Carotid IMT, which is the distance between the lumen–intima interface and media–adventitia interface, was defined as the mean of the three measurements on both CCAs. Increased cIMT was defined as an increase in mean cIMT by ≥1.0 mm [28]. A carotid plaque was defined as a cIMT ≥ 1.5 mm and the presence of focal protrusion into the arterial lumen that is at least 50% larger compared to the adjacent lumen [29]. Carotid atherosclerosis was defined by increased cIMT and presence of carotid plaque.

### 2.6. Statistical Analysis 

All continuous data are expressed as medians with interquartile ranges and were compared using the Mann–Whitney test after undergoing a test for normality. Categorical data are expressed as numbers and percentages and were compared using the χ-squared test or Fisher’s exact test. The association between LSMM and subclinical atherosclerosis, including increased cIMT and carotid plaques, was assessed using logistic regression analysis with backward elimination. Several models with sequential adjustment for confounding variables were analyzed using multivariate logistic regression analysis. Statistical significance was defined as *p* < 0.05. All statistical analyses were performed using the R software (version 3.0.2; R Foundation for Statistical Computing, Vienna, Austria. R 3.0.2 is released (26 September 2013)).

## 3. Results

### 3.1. Baseline Characteristics 

Table 1 shows the baseline characteristics of the participants as stratified by the presence or absence of LSMM_BMI. Of the 683 patients with NAFLD who underwent carotid ultrasound, 75 (11.0%) were diagnosed with LSMM. When compared with patients without LSMM (non-LSMM group), those with LSMM (LSMM group) were older and had more negative metabolic characteristics and comorbidities including high BMI, larger WC, obesity, hypertension, and metabolic syndrome. The levels of AST, ALT, insulin, and HOMA-IR were significantly higher in the LSMM group than in the non-LSMM group. The median cIMT was significantly higher in the LSMM group compared to the non-LSMM group (0.85 vs. 0.75 mm, *p* <0.001). The proportions of increased cIMT and carotid plaques were higher in the LSMM group than in the non-LSMM group (33.3% vs. 14.5% for increased cIMT and 20.0% vs. 8.1% for carotid plaques; *p* = 0.001; Table 1, Figure 2).

### 3.2. Association of LSMM with Carotid Atherosclerosis in Patients with NAFLD 

To evaluate the association of LSMM with carotid atherosclerosis in patients with NAFLD, we performed multivariate analysis using adjusted models. After adjusting for age and sex, LSMM was found to be independently associated with increased cIMT (odds ratio (OR), 2.26; 95% confidence interval (CI), 1.26–4.04; *p* = 0.006). These results were consistent after further adjustment for DM and hypertension (Model 1: OR, 2.28; 95% CI, 1.27–4.08; *p* = 0.005); for obesity, WC, and hyperuricemia (Model 2: OR, 2.28; 95% CI, 1.27–4.08; *p* = 0.005); and for lipid profile, AST, ALT, GGT, HOMA-IR, and hsCRP (Model 3: OR, 2.26; 95% CI, 1.26–4.04; *p* = 0.006). After adjusting for age and sex, LSMM was found to be independently associated with the presence of carotid plaques (OR, 2.05; 95% CI, 1.03–4.08; *p* = 0.004). These results were consistent after further adjustment for DM and hypertension (Model 1: OR, 2.20; 95% CI, 1.10–4.40; *p* = 0.026); for obesity, WC, and hyperuricemia (Model 2: OR, 2.90; 95% CI, 1.40–6.04; *p* = 0.004); and for lipid profile, AST, ALT, GGT, HOMA-IR, and hsCRP (Model 3: OR, 2.74; 95% CI, 1.30–5.78; *p* = 0.008) (Table 2).

LSMM_BMI was adopted by the Foundation for National Institutes of Health as a definition of sarcopenia, which was defined as <0.789 in men or <0.512 in women.

Model 1: Age, sex, presence of diabetes mellitus and hypertension.

Model 2: Further adjusted for presence of obesity, waist circumference, and hyperuricemia.

Model 3: Further adjusted for total cholesterol, triglyceride, high-density lipoprotein, alanine aminotransferase, aspartate aminotransferase, gamma-glutamyl transferase, homeostatic model assessment of insulin resistance, and high-sensitivity C-reactive protein.

### 3.3. Association of LSMM with Carotid Atherosclerosis in Obese Patients with NAFLD 

We compared the prevalence of carotid atherosclerosis based on obesity and LSMM. The proportions of increased cIMT and carotid plaques were higher in the obese/LSMM group than in the obese/non-LSMM group (33.3% vs. 12.7% for increased cIMT and 17.6% vs. 5.7% for carotid plaques, *p* = 0.001); this increase was absent in the non-obese group regardless of the LSMM. In the subgroup analysis of the obese group, the proportion of carotid atherosclerosis was higher in patients with LSMM than in those without LSMM. (33.3% vs. 17.6% for increased cIMT and 12.7% vs. 5.7% for carotid plaques, *p* < 0.001; Figure 3). However, in the subgroup analysis of the non-obese group, the proportion of carotid atherosclerosis was not significantly different between patients with LSMM and those without LSMM (42.9% vs. 18.0% for increased cIMT, *p* = 0.247 and 42.9% vs. 12.7% for carotid plaques, *p* = 0.084; data not shown).

To evaluate the association of LSMM with carotid atherosclerosis in obese patients with NAFLD, we performed multivariate-adjusted analysis as previously described. After adjusting for age and sex, LSMM was found to be independently associated with increased cIMT (OR, 2.44; 95% CI, 1.29–4.60; *p* = 0.006). These results were maintained after further adjustment for DM and hypertension (Model 1: OR, 2.44; 95% CI, 1.29–4.60; *p* = 0.006); for WC and hyperuricemia (Model 2: OR, 2.45; 95% CI, 1.30–4.57; *p* = 0.006); and for lipid profile, AST, ALT, GGT, HOMA-IR, and hsCRP (Model 3: OR, 2.44; 95% CI, 1.29–4.60; *p* = 0.006). After adjusting for age and sex, LSMM was found to be independently associated with the presence of carotid plaques (OR, 2.56; 95% CI, 1.15–5.71; *p* = 0.022). These results were maintained after further adjustment for DM and hypertension (Model 1: OR, 2.91; 95% CI, 1.28–6.60; *p* = 0.011); for WC and hyperuricemia (Model 2: OR, 2.91; 95% CI, 1.28–6.60; *p* = 0.011); and for lipid profile, AST, ALT, GGT, HOMA-IR, and hsCRP (Model 3: OR, 2.70; 95% CI, 1.18–6.16; *p* = 0.019) (Table 3).

LSMM_BMI was adopted by the Foundation for National Institutes of Health as a definition of sarcopenia, which was defined as <0.789 in men or <0.512 in women.

Model 1: Age, sex, presence of diabetes mellitus and hypertension

Model 2: Further adjusted for waist circumference and hyperuricemia

Model 3: Further adjusted for total cholesterol, triglyceride, high-density lipoprotein, alanine aminotransferase, aspartate aminotransferase, gamma-glutamyl transferase, homeostatic model assessment of insulin resistance, and high-sensitivity C-reactive protein.

## 4. Discussion

In our study, we demonstrated that LSMM is associated with an increased risk of developing carotid atherosclerosis, including cIMT and carotid plaques, independent of traditional metabolic factors and IR. Using stepwise adjustment, we demonstrated an approximately 2-fold increase in adjusted risk of LSMM for carotid atherosclerosis in patients with NAFLD. In addition, we demonstrated an approximately 2.5-fold increase in adjusted risk of LSMM for carotid atherosclerosis in obese patients with NAFLD.

Classically, carotid atherosclerosis is a strong predictive marker for future CVD events as defined by an increase in cIMT. The presence of carotid plaques can be measured in a noninvasive manner using carotid ultrasound [15,30]. In a previous study, increased cIMT was correlated with the risk of CVD events in patients with DM but not in those without DM [31].

Sarcopenia is regarded as a potential risk factor for developing NAFLD with or without advanced fibrosis as the two have common putative mechanisms, one of which is chronic inflammation [17,18,26,32]. Recently, Seo et al. and Campos et al. demonstrated that low muscle mass is associated with subclinical atherosclerosis in patients with DM and the old population, respectively [15,33]. In one Korean study that grouped patients according to skeletal muscle mass quartiles and stratified them by sex, carotid atherosclerosis had an adjusted OR that was approximately 2-fold greater in patients with concurrent DM regardless of sex [15]. Different imaging modalities, such as cardiac computed tomography with coronary calcium score, have been used to assess subclinical atherosclerosis; Campos et. al. demonstrated an association between sarcopenia and subclinical atherosclerosis in very elderly individuals [33].

While the pathophysiological mechanisms linking LSMM and the development of carotid atherosclerosis in patients with NAFLD are not clearly understood, the putative mechanisms are as follows: Insulin resistance (IR) and chronic inflammation have been determined to be the common mechanism between NAFLD and LSMM in previous studies [11,34]. Considering that skeletal muscle is an insulin-sensitive organ, loss of skeletal muscle mass exacerbates IR and glycogenolysis, thereby leading to lipolysis in adipose tissue and excess free fatty acid (FFA) levels [35]. Persistent hyperglycemia and excess FFA levels produce cytokines, including IL-6 and nuclear factor κB (NF-κB), which lead to chronic systemic inflammation [32]. Eventually, chronic inflammation with worsening IR exacerbates proteolysis, lipolysis, and gluconeogenesis in a vicious cycle [32].

In vascular endothelial cells, excess oxidative stress and inflammatory cytokines, such as NF-κB, induce the release of chemotactic factors, adhesion molecules, and pro-inflammatory cytokines, leading to the promotion of monocyte recruitment and endothelial dysfunction [36,37]. Progressive endothelial injury exacerbates carotid atherosclerosis, increasing the risk of CVD and mortality [2]. In previous studies, an increased risk of CVD was associated with VHD and arrhythmia in patients with NAFLD [5,6,7].

The strengths of this study are as follows: First, to the best of our knowledge, this is the first well-characterized cohort study to evaluate the association between LSMM and carotid atherosclerosis in patients with NAFLD. The definitions of LSMM and NAFLD were taken from previous publications [4,26]. Second, we investigated LSMM as an independent risk factor for carotid atherosclerosis using stepwise adjustment for classic metabolic factors in patients with NAFLD as well as in patients with both NAFLD and obesity.

Careful interpretation is needed due to several limitations in our study. First, because this is a cross-sectional, single-center, small-sized, retrospective study, the causal association between LSMM and carotid atherosclerosis is uncertain. Further multicenter, prospective, longitudinal, large-cohort studies are required to evaluate the causal association between these diseases. Second, it is difficult to generalize the results of this study to the entire population due to the potential for selection bias for subjects who are concerned about their own health and have the ability to pay for medical care. However, similar to one study involving the entire Korean population in which the proportion of sarcopenia in patients with NAFLD was approximately 12%, the proportion of LSMM in our study was 11.0% [32]. The consistency between these results suggests that this study may have a low potential for selection bias. Third, patient history of smoking, medication intake for metabolic disorders, vitamin D level, and dietary/lifestyle habits were not measured due to inadequate retrospective data. Considering the importance of dietary/lifestyle modification in NAFLD patients, further studies are warranted to evaluate the association between dietary/lifestyle modification and the progression of sarcopenia. Fourth, it was difficult to reveal an association between LSMM and the non-NAFLD group due to inadequate data. Fifth, the muscle function as a prerequisite of sarcopenia and performance status of the subjects were not evaluated. However, BIA is a safe and simple modality for assessing SMM that does not involve radiation exposure; thus, it can be easily utilized in clinical practice. It has also been well validated in large population-based studies [4,26]. Further studies associated with sarcopenia are needed to investigate muscle function with muscle quantity and/or muscle performance status.

In conclusion, LSMM is associated with an increased risk of carotid atherosclerosis in patients with NAFLD independent of metabolic risk factors. Considering that carotid atherosclerosis is a predictor of future CVD events, physicians need to assess the presence of carotid atherosclerosis, as well as SMM, in patients with NAFLD. Further prospective, longitudinal studies are warranted to confirm the causal association between LSMM and subclinical atherosclerosis in patients with NAFLD.

## Figures and Tables

**Figure 1 diagnostics-11-00854-f001:**
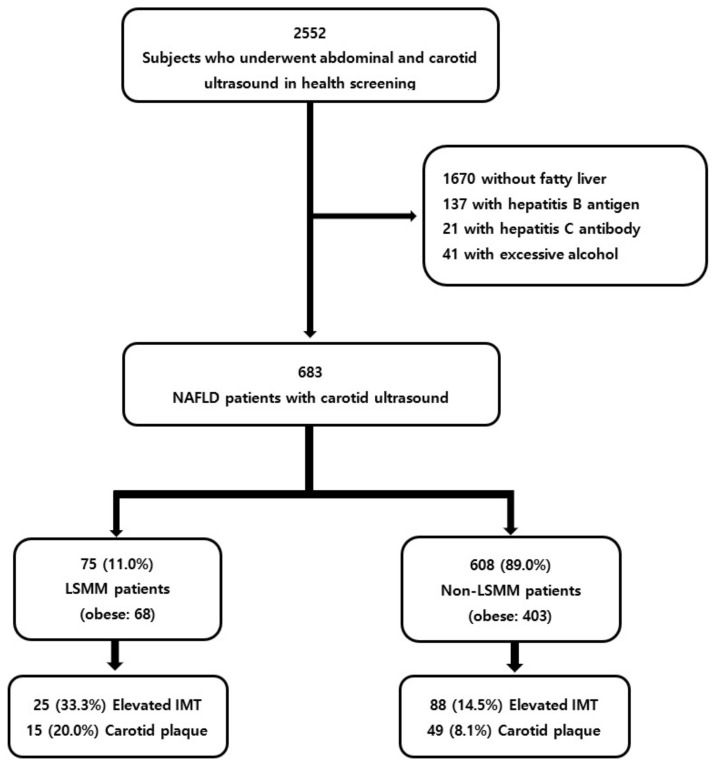
Flow chart of the enrolled patients.

**Figure 2 diagnostics-11-00854-f002:**
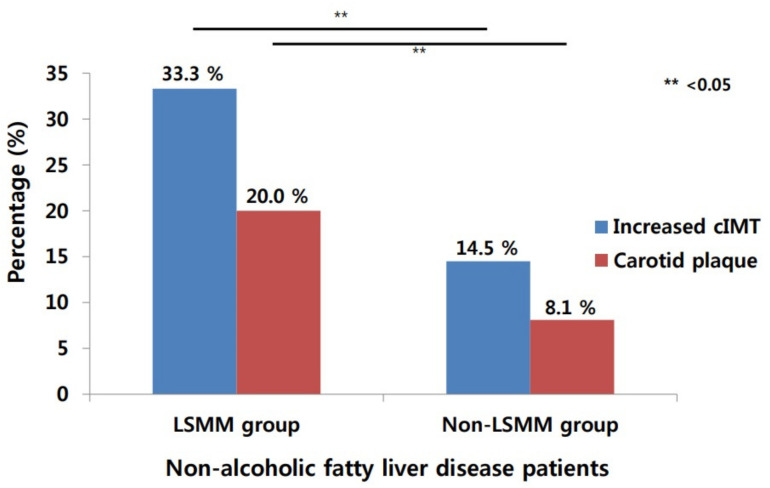
Percentage of increased intima–media thickness and carotid plaque according to presence or absence of LSMM in patients with nonalcoholic fatty liver disease. cIMT, carotid intima–media thickness; LSMM, low skeletal muscle mass.

**Figure 3 diagnostics-11-00854-f003:**
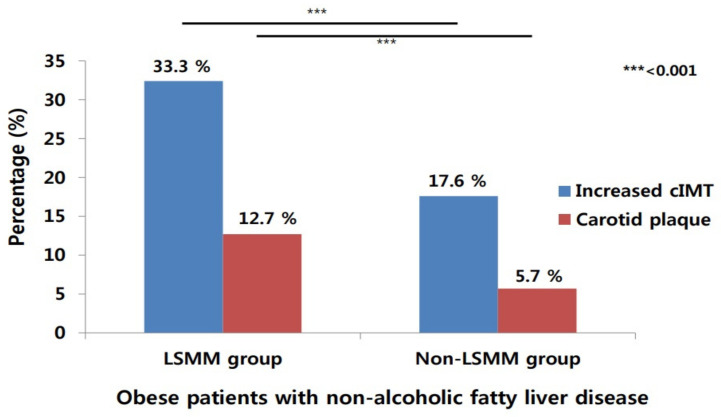
Percentage of elevated intima–media thickness and carotid plaque according to presence of LSMM in obese patients with nonalcoholic fatty liver disease. cIMT, carotid intima–media thickness; LSMM, low skeletal muscle mass.

**Table 1 diagnostics-11-00854-t001:** Baseline characteristics of the study population.

Variable	LSMM Group *n* = 75 (11.0%)	Non-LSMM Group *n* = 608 (89.0%)	*p*-Value *
Age, years	53.0 [45.0–62.0]	49.0 [42.0–55.0]	0.005
Male, *n* (%)	66 (88.0)	522 (85.9)	0.742
BMI, kg/m^2^	28.9 [26.6–30.6]	26.1 [24.4–27.9]	<0.001
Waist circumference, cm	90.0 [86.5–96.5]	87.0 [83.0–92.0]	<0.001
Comorbidities, *n* (%)			
Obesity	68 (90.7)	403 (66.3)	<0.001
Diabetes mellitus	13 (17.3)	91 (15.0)	0.713
Hypertension	35 (46.7)	164 (27.0)	0.001
Metabolic syndrome	46 (61.3)	252 (41.4)	0.002
Liver function profiles			
Aspartate aminotransferase, IU/L	30.0 [24.0–38.5]	27.0 [23.0–34.0]	0.020
Alanine aminotransferase, IU/L	41.0 [29.5–58.0]	33.0 [24.0–48.0]	0.002
Platelet count, K/uL	241.0 [206.0–281.5]	239.5 [207.0–278.0]	0.892
Gamma-glutamyl transferase, IU/L	39.0 [30.0–49.5]	35.0 [24.0–52.0]	0.096
Albumin, g/dL	4.8 [4.6–5.0]	4.8 [4.6–5.0]	0.365
Glucose profiles			
Fasting plasma glucose, mg/dL	103.0 [93.0–113.0]	98.0 [90.0–108.0]	0.043
Insulin level, microU/mL	10.0 [7.7–14.2]	8.1 [6.0–11.1]	<0.001
HOMA-IR	2.6 [2.0–3.8]	2.0 [1.4–2.9]	<0.001
Lipid profiles			
Total cholesterol, mg/dL	211.0 [186.5–245.0]	206.0 [179.0–230.0]	0.167
Triglyceride, mg/dL	165.0 [126.5–230.0]	155.5 [110.0–216.0]	0.301
High-density lipoprotein, mg/dL	49.1 [40.9–55.8]	48.7 [42.1–56.5]	0.801
hsCRP, mg/dL	0.09 [0.05–0.16]	0.08 [0.05–0.16]	0.279
Median carotid IMT, mm	0.85 [0.73–1.05]	0.75 [0.64–0.85]	<0.001
Increased cIMT, *n* (%)	25 (33.3)	88 (14.5)	<0.001
Carotid plaque, *n* (%)	15 (20.0)	49 (8.1)	0.002

Data are expressed as median with interquartile range (IQR) or numbers (%). * Calculated by the Mann–Whitney U test after a normality test and chi-squared test (or Fisher’s exact test, if appropriate). LSMM_BMI was adopted by the Foundation for National Institutes of Health as a definition of sarcopenia, which was defined as <0.789 in men or <0.512 in women. LSMM, low skeletal muscle mass; BMI, body mass index; HOMA-IR, homeostasis model of insulin resistance; hsCRP, high-sensitivity C-reactive protein; cIMT, carotid intima–media thickness.

**Table 2 diagnostics-11-00854-t002:** Adjusted odds ratios of low skeletal muscle mass for increased intima–media thickness and carotid plaque in patients with nonalcoholic fatty liver disease.

	Increased cIMT		Carotid Plaque	
	OR (95% CI)	*p*-Value	OR (95% CI)	*p*-Value
**LSMM_BMI in NAFLD Patients (yes vs. no)**				
Unadjusted	2.95 (1.74–5.02)	<0.001	2.85 (1.51–5.39)	0.001
Age, sex adjusted	2.26 (1.26–4.04)	0.006	2.05 (1.03–4.08)	0.004
Model 1	2.28 (1.27–4.08)	0.005	2.20 (1.10–4.40)	0.026
Model 2	2.28 (1.27–4.08)	0.005	2.90 (1.40–6.04)	0.004
Model 3	2.26 (1.26–4.04)	0.006	2.74 (1.30–5.78)	0.008

cIMT, carotid intima–media thickness; OR, odds ratio; CI, confidence interval; LSMM, low skeletal muscle mass; BMI, body mass index; NAFLD, nonalcoholic fatty liver disease.

**Table 3 diagnostics-11-00854-t003:** Adjusted odds ratios of low skeletal muscle mass for elevated intima–media thickness and carotid plaque in obese patients with nonalcoholic fatty liver disease.

	Increased cIMT		Carotid Plaque	
	OR (95% CI)	*p*-Value	OR (95% CI)	*p*-Value
**LSMM_BMI in Obese NAFLD Patients (yes vs. no)**				
Unadjusted	3.30 (1.84–5.94)	<0.001	3.54 (1.67–7.51)	0.001
Age, sex adjusted	2.44 (1.29–4.60)	0.006	2.56 (1.15–5.71)	0.022
Model 1	2.44 (1.29–4.60)	0.006	2.91 (1.28–6.60)	0.011
Model 2	2.45 (1.30–4.57)	0.006	2.91 (1.28–6.60)	0.011
Model 3	2.44 (1.29–4.60)	0.006	2.70 (1.18–6.16)	0.019

cIMT, carotid intima–media thickness; OR, odds ratio; CI, confidence interval; LSMM, low skeletal muscle mass; BMI, body mass index; NAFLD, nonalcoholic fatty liver disease.

## Data Availability

The data used to support the findings of this study are available from the corresponding author upon request.

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
