# Peer review of "Low Skeletal Muscle Mass Is a Risk Factor for Subclinical Atherosclerosis in Patients with Nonalcoholic Fatty Liver Disease"

_diagnostics, 2021, doi:10.3390/diagnostics11050854_

Round 1

Reviewer 1 Report

  1. This is a nicely written manuscript on an important topic, NAFLD.
  2. What did the “one study” on line 52 find, and how is this investigation novel and set to advance the field? This should be described in the introduction and lead to a directional hypothesis statement.
  3. Inclusion of a biostatistician would provide rigor to this study.
  4. Are these individuals technically obese with mean BMI = 28? It needs to be included in the flow chart in figure 1 the number of obese patients used.
  5. What about comparison of obese vs. non-obese for cIMT and carotid plaques in NAFLD patients?
  6. It is not exactly clear what “LSMM_BMI” and “Non-LSMM_BMI” mean in table 1?
  7. What are results in patients without NAFLD but with LSMM?
  8. Do any of the patients have clinically defined sarcopenia? The rationale in the introduction is based on sarcopenia.
  9. Metabolic factors do not seem to be the mechanism linking LSMM and CVD. But, in the discussion, it is mentioned that insulin resistance-mediated inflammation is a mechanism. Were insulin levels associated with anything here? Also, CRP levels were not different between here, are there any other inflammatory factors that should be looked at regarding this mechanistic link?

Author Response

Thank you very much for the meticulous review.

The response to the reviewer's comment has been summarized and attached as a word file. Please see the attachment.

Reviewer 2 Report

To the authors of the manuscript,

I have found the association between atherosclerosis and sarcopenia. After revising the work, I recommend to accept it with MINOR CHANGES.

My main concern is that the possible relationship between dietary or lifestyle habits and sarcopenia development. At least it should be included in the discussion.

Author Response

(The authors gave the same response as above.)

Round 2

Reviewer 1 Report

The authors have adequately addressed my previous concerns.